# The tetracycline resistome is shaped by selection for specific resistance mechanisms by each antibiotic generation

Kevin S. Blake [1,2], Yao-Peng Xue[1], Vincent J. Gillespie[1], Skye R. S. Fishbein [1,2], Niraj H. Tolia [3] ✉, Timothy A. Wencewicz [4] ✉ & Gautam Dantas [1,2,5,6,7] ✉

The history of clinical resistance to tetracycline antibiotics is characterized by cycles whereby the deployment of a new generation of drug molecules is quickly followed by the discovery of a new mechanism of resistance. This suggests mechanism-specific selection by each tetracycline generation; however, the evolutionary dynamics of this remain unclear. Here, we evaluate 24 recombinant *Escherichia coli* strains expressing tetracycline resistance genes from each mechanism (efflux pumps, ribosomal protection proteins, and enzymatic inactivation) in the context of each tetracycline generation. We employ a high-throughput barcode sequencing protocol that can discriminate between strains in mixed culture and quantify their relative abundances. We find that each mechanism is preferentially selected for by specific antibiotic generations, leading to their expansion. Remarkably, the minimum inhibitory concentration associated with individual genes is secondary to resistance mechanism for inter-mechanism relative fitness, but it does explain intra-mechanism relative fitness. These patterns match the history of clinical deployment of tetracycline drugs and resistance discovery in pathogens.

An organism's fitness relative to other organisms is a product of its present environment, and thus may increase or decrease under different environmental conditions[1,2]. Natural selection balances the benefits of encoding an expansive repertoire of special functions that enable an organism to survive rare perturbations, with the metabolic costs of maintaining those functions. Antibiotic resistance in bacteria is an extreme example of this trade-off, as bacterial cells lacking resistance during antibiotic exposure suffer immediate consequences (i.e., inhibited growth or cell death), but can acquire and eject antibiotic resistance genes with relative ease[3,4]. Investigating the phenotypic and genotypic evolution of antibiotic resistance is of interest due

to the worldwide increase in antibiotic resistance in human pathogens[5,6], and also because it can be used to experimentally investigate fundamental evolutionary dynamics[7–9].

Resistance to specific antibiotic classes can be achieved by several genetically encoded mechanisms[10]. These are generalized into three categories: (1) keeping the intracellular antibiotic concentration low (e.g., permeability barrier, drug efflux), (2) modifying the antibiotic target (e.g., mutation, overexpression, protection proteins), and (3) inactivating the antibiotic molecule (e.g., enzymatic degradation, modification)[11]. The magnitude of benefits and costs associated with resistance in an organism can differ substantially between and within

[1]The Edison Family Center for Genome Sciences and Systems Biology, Washington University School of Medicine, St. Louis, MO, USA. [2]Department of Pathology and Immunology, Division of Laboratory and Genomic Medicine, Washington University School of Medicine, St. Louis, MO, USA. [3]Host-Pathogen Interactions and Structural Vaccinology Section, Laboratory of Malaria Immunology and Vaccinology, National Institute of Allergy and Infectious Diseases, National Institutes of Health, Bethesda, MD, USA. [4]Department of Chemistry, Washington University in St. Louis, St. Louis, MO, USA. [5]Department of Molecular Microbiology, Washington University School of Medicine, St. Louis, MO, USA. [6]Department of Biomedical Engineering, Washington University in St. Louis, St. Louis, MO, USA. [7]Department of Pediatrics, Washington University School of Medicine, St. Louis, MO, USA. ✉e-mail: niraj.tolia@nih.gov; wencewicz@wustl.edu; dantas@wustl.edu

mechanisms, but the relative growth rate associated with an antibiotic resistance gene is often negatively correlated with the fold-increase in minimum inhibitory concentration (MIC)[12]. Determining potential trade-offs is important for understanding which resistance mechanisms will transfer to and persist within human pathogens[13].

Since their discovery in the 1940s, the tetracycline family of antibiotics has been intensively used in agriculture and the clinic[14–16]. These drugs inhibit bacterial protein synthesis by binding to the 16S rRNA of the 30S ribosome subunit, preventing accommodation of incoming aminoacyl-tRNAs[17]. Tetracyclines are type II polyketides composed of a four (tetra-) ring (-cycl-) scaffold[18]. First-generation tetracyclines include the naturally occuring chlortetracycline (1948), oxytetracycline (1950), and tetracycline (1953)[19,20]. Over the decades, to overcome resistance to existing tetracyclines, the core tetracycline scaffold has been modified with a variety of functional groups in an effort to improve ADME-Tox (absorption, distribution, metabolism, excretion, and toxicity), efficacy, and spectrum of activity[21]. These semi-synthetic and fully synthetic derivatives are referred to as second- and third-generation tetracycines, respectively. The second-generation tetracyclines include minocycline (1961), metacycline (1962), and doxycycline (1967)[22]. Third-generation tetracyclines (i.e., glycylcyclines) are the most recently developed and include tigecycline (reported 1993, FDA-approved 2005), omadacycline (reported 2013, FDA-approved 2018) and eravacycline (reported 2013, FDA-approved 2018)[18,23]. These later-generation drugs were intentionally optimized for activity against multidrug-resistant pathogens[24,25], and tigecycline is currently reserved as an antibiotic of last resort[26–29].

The history of tetracycline resistance in human pathogens has been characterized by cycles where new generations of tetracycline antibiotics are deployed, quickly followed by the discovery of new mechanisms of resistance (Fig. 1a). Following the discovery of first-generation tetracyclines in 1948, the first tetracycline-resistant bacterium, *Shigella dysenterieae*, was reported in 1953 and the first multidrug-resistant *Shigella* in 1955[30–33]. These pathogens were later determined to encode tetracycline efflux pumps (EFF)[30]. Tetracycline EFFs are membrane-associated proteins belonging to the major facilitator superfamily[34] (Fig. 1b). By exporting tetracyclines from the cell EFFs reduce intracellular tetracycline concentration, thereby protecting ribosomes from tetracycline binding. EFFs have historically been the most numerous tetracycline-resistance mechanism, representing 60% of all *tet* and *otr* genes in 2005[35], and continue to be so today with at least 33 characterized determinants[21].

Pathogen outbreaks resistant to first-generation drugs through the expression of EFFs motivated the development of second-generation tetraycline in the 1960s. These drugs had improved activity against strains expressing EFFs, presumably due to increased uptake imparted by the C7-dimethylamino group[36,37]. The first tetracycline resistance-conferring ribosomal protection proteins (RPP) were reported shortly thereafter, in clinical isolates of *Streptococcus agalactiae* in 1980[38,39]. RPPs are translational GTPases with homology to the elongation factor EF-G[40]. RPPs protect the ribosome by binding to the 30S and 50S subunits, sterically occluding and distorting the tetracycline binding site[41] (Fig. 1b). To date there are at least 13 characterized RPP genes[21,35].

Lastly, the third-generation tetracyclines, introduced from 1993–2013, included a bulky glycylamine side chain at the C9-position of the D-ring strategically placed to sterically occlude RPP-binding and prevent chasing from the drug binding site[24,25]. Their deployment was soon followed by the discovery of tetracycline destructases (TDase). In 2013 these tetracycline-inactivating enzymes were identified in clinical isolates from a hospital in Sierra Leone[42–44]. TDases are class A flavin-dependent monooxygenases that covalently modify and inactivate

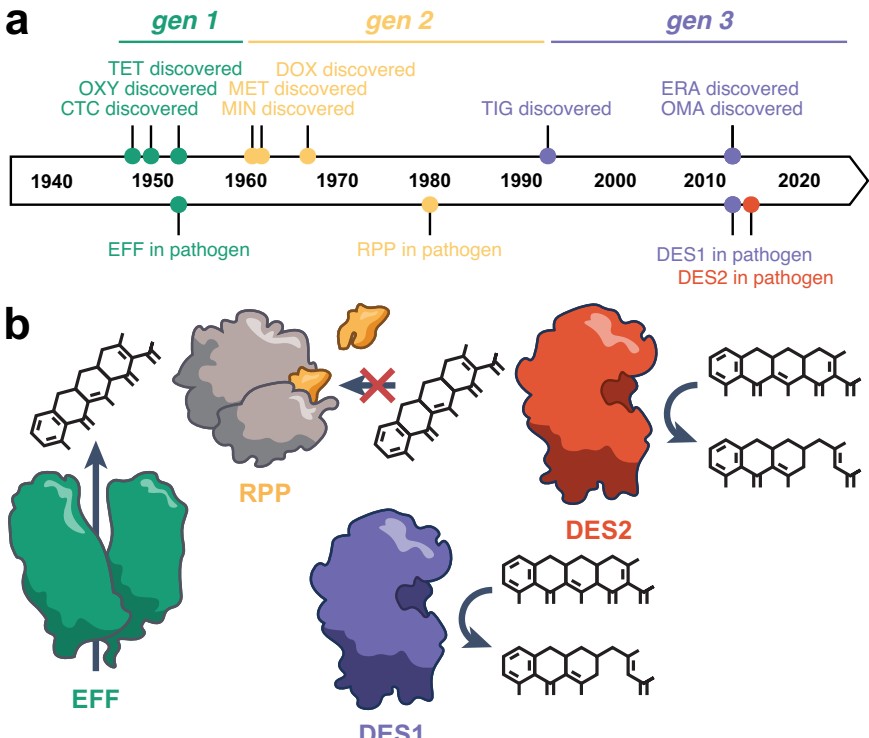

**Fig. 1 | The clinical deployment of new generations of tetracyclines has been quickly followed by the discovery of new mechanisms of resistance. a** Timeline of the discovery of first-generation, second-generation, and third-generation tetracycline antibiotics (top), and the first discovery of each mechanism of tetracycline resistance in human pathogens (bottom). **b** Cartoon representation of each mechanism of tetracycline resistance. (Not drawn to scale.) TET tetracycline, DOX doxycycline, CTC chlortetracycline, DOX doxycycline, MIN minocycline, MET metacycline, TIG tigecycline, OMA omadacycline, ERA eravacycline, EFF efflux pump, RPP ribosomal protection protein, DES1 type 1 tetracycline destructase, DES2 type 2 tetracycline destructase.

tetracycline antibiotics via hydroxylation of C11a and/or oxygen insertion at C1[45,46]. It has not been empirically determined whether or not tetracycline-inactivating enzymes are secreted; however, their requirement for NADPH to reduce the flavin cofactor during enzyme turnover likely means they are only functional in the cytoplasm[21]. Because only two TDase enzymes were known prior to 2015−Tet(X) and Tet(X2), which differ by just one amino acid−it had been presumed that these enzymes were insignificant clinically[47]. However, that paradigm has been upended in recent years with a flurry of reports describing new TDase sequences, beginning in 2015 with the discovery of 10 new enzymes[45] and continuing in 2019-2020 with reports totaling over 20 new TDases discovered in environmental and pathogenic bacteria from diverse habitats[48–55]. Based on sequence identity, TDases are divided into two clades: type 1 destructases (DES1) found in many human pathogens and human gut microbiomes, and type 2 tetracycline destructases (DES2) found predominately in soil metagenomes and some *Legionella* species[45,56] (Fig. 1b). While both types can confer resistance to first- and second-generation tetracyclines, DES1s can confer resistance to third-generation drugs while DES2s cannot[54]. Because of this important difference, for the purposes of this report we treat DES1 and DES2 as separate mechanisms. If tetracycline gene fitness is judged solely by the breadth of drugs to which the gene confers resistance, DES1s would be the most fit as neither EFFs nor RPPs have demonstrated resistance to third-generation tetracyclines.

There has been a marked increase in tetracycline-resistance gene abundance, diversity, and taxonomic breadth over time, in association with increased production and use of tetracycline antibiotics (Supplementary Fig. 1a)[18,30,35,57]. In a retrospective analysis of archival soils dating back to the 1940s, tetracycline-resistance genes had the highest rates of increased abundance compared to other major drug classes[57]. Yet, tetracycline resistance in bacterial populations predates the modern selective pressure of human production by at least 30,000 years[58], indicating that the emergence of resistance in pathogens is the result of selection from a natural reservoir of tetracycline-resistance genes.

Here, we investigated the historical pattern whereby the emergence of a new mechanism of tetracycline resistance in pathogens coincides with the clinical deployment of a new generation of tetracycline antibiotics. In particular, we were interested in the question of why RPPs or DES1s did not appear in pathogens until >30-60 years after the deployment of first-generation tetracyclines and the discovery of EFFs, despite all mechanisms conferring similar degrees of resistance to first-generation drugs. We hypothesized that each mechanism has a selective advantage relative to the others in the context of the drug generation deployed immediately prior to their emergence; first-generation tetracyclines select for EFFs, second-generations for RPPs, and third-generations for DES1. In other words, tetracycline-resistance genes are not functionally redundant but specialized to each tetracycline antibiotic generation. This would explain the historical pattern of emergence, as when a bacterial reservoir encoding multiple tetracycline-resistance mechanisms is exposed to a tetracycline antibiotic, different mechanisms will be preferentially enriched depending on which generation the drug belongs to. To investigate this dynamic we evaluated 24 tetracycline-resistance genes (6 EFFs, 6 RPPs, 6 DES1s, and 6 DES2s) by heterologous expression in *Escherichia coli* and quantified these genes' phenotypic benefits (increased MIC) and associated costs (decreased growth rate) in monoculture. Then, to directly evaluate each strain's relative fitness and determine if different mechanisms have a selective advantage in the context of specific drug generations, we grew them in mixed-culture competition assays and quantified their changes in relative abundance.

## Results

### Resistance gene benefits and costs cluster by resistance mechanism

We constructed an isogenic library of 24 *E. coli* DH5αZ1 strains, each encoding a unique tetracycline-resistance gene. An isogenic system was designed to control for confounding variables, such that any phenotypic differences between strains can be attributable to the resistance genes. We selected *E. coli* as the host because wild isolates have been observed encoding several tetracycline-resistance genes, and lab strains are commonly used to evaluate these genes heterologously[56,59,60]. Six genes from each of the four mechanisms of tetracycline resistance were selected (Supplementary Data 1): efflux pumps (EFF), ribosomal protection proteins (RPP), type 1 destructases (DES1), and type 2 destructases (DES2). We selected genes that were most preponderant in the literature and represented each mechanism's overall sequence diversity (Supplementary Fig. 1b). Selected genes have among the highest taxonomic diversity from each mechanism, with many identified in both Gram-positive and Gram-negative taxa (Supplementary Fig. 1c; Supplementary Data 2). Of the 583 genera in which tetracycline-resistance genes were identified, 339 are represented by the selected genes (Supplementary Fig. 1d). Wild-type versions of these genes plus a gene-specific 7 base pair DNA barcode were inserted into the pZE24 plasmid (Supplementary Fig. 2a). For a negative control, we also prepared a plasmid that did not contain a tetracycline-resistance gene (i.e., empty vector) but did have a barcode. These plasmid constructs were then transformed into *E. coli* DH5αZ1, enabling inducible gene expression with the addition of IPTG (Supplementary Fig. 2b).

To evaluate the benefits provided by these tetracycline-resistance genes in the context of the same *E. coli* host, we performed antibiotic susceptibility testing with nine tetracycline antibiotics, three from each generation (Fig. 2a; Supplementary Fig. 3; Supplementary Data 3). Genes from all mechanisms conferred resistance to first- and second-generation tetracyclines; however, as previously reported[54,56], only DES1 genes conferred resistance to third-generation tetracyclines. We found that the minimum inhibitory concentrations (MIC) for these strains against all drugs generally clustered by resistance mechanism. However, when MIC profiles were evaluated with just first-generation tetracyclines, or first- and second-generation tetracyclines−simulating the history of tetracycline use before the discovery of later-generation tetracyclines−this clustering broke down (Supplementary Fig. 4). In both instances, the EFF and DES1 strains were intermixed, indicating similar MIC profiles.

We next quantified the potential costs that expression of these resistance genes incur on their *E coli* host. We measured the maximal growth rate of these strains with full expression of their tetracycline-resistance gene but in the absence of tetracycline. As expected, the empty vector strain not incurring the metabolic costs of expressing a resistance gene had among the highest growth rate (Fig. 2b). The growth rates of 6/6 EFF strains and 5/6 DES2 strains were not significantly lower than the empty vector strain (one-way ANOVA FDR $q > 0.05$). In contrast, 4/6 RPP strains and 5/6 DES1 strains had significantly lower growth rates (one-way ANOVA FDR $q < 0.05$). When the average growth rates of all strains per mechanism were binned together, the RPP and DES1 bins had significantly lower growth rates compared to the empty strain (one-way ANOVA FDR $q < 0.05$; Fig. 2c). These findings show that the DES1 and RPP genes incur significant costs in terms of growth rate, while EFF and DES2 do not.

By combining our MIC and growth rate data, we identified patterns of benefits and costs (Fig. 2d). We observed that in the context of first-generation drugs, EFF strains had both the highest MICs and no significant reduction in growth rate. Although the DES1 and RPP genes confer similarly high levels of resistance to these drugs they also are associated with significant growth defects. DES2s did not incur a growth defect but conferred lower MICs than EFFs. In a mixed

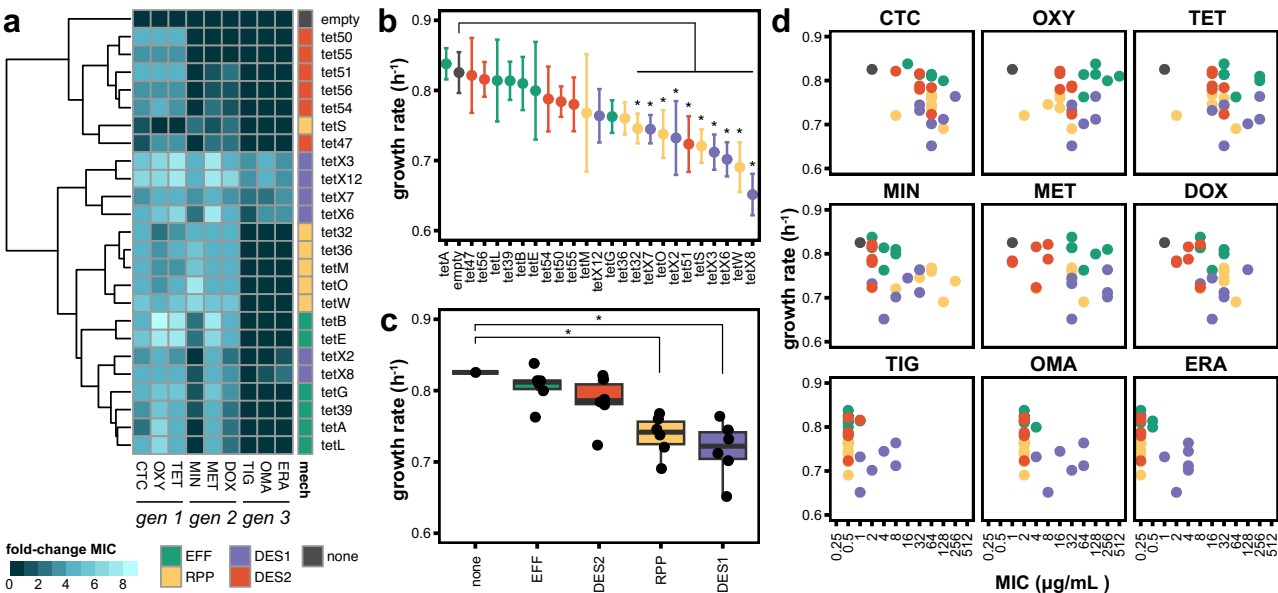

**Fig. 2 | Mechanism-specific differences in benefits and costs associated with tetracycline-resistance genes in monoculture. a** Heatmap of AST performed for each strain against tetracycline antibiotics from each generation. The resulting MIC values for each strain-antibiotic combination are plotted as log2 fold-change over the empty vector strain's MIC for that antibiotic. MIC values are listed in Supplementary Data 4. Rows are ordered by a hierarchical grouping of MIC values; columns are ordered by drug discovery date. **b** Maximal growth rates measured for each strain grown in tetracycline-free media with full gene expression using IPTG. The maximal growth rate was calculated by applying a rolling regression with a sliding window of 1 h to log-transformed $OD_{600}$ growth curves. Points represent the average of three technical replicates, and error bars represent the standard deviation. Asterix indicates FDR $q$-value < 0.05 using ordinary one-way ANOVA with Benjamini and Hochberg. Colored by mechanism. **c** Boxplots of growth rates binned by mechanism. Points represent the average of three technical replicates. Asterix indicates FDR $q$-value < 0.05 using ordinary one-way ANOVA with Benjamini and Hochberg. Boxes show median and quartiles; error bars extend to the values within the 1.5 interquartile range. **d** Merged MIC and growth rate data for each strain-antibiotic combination. Colored by mechanism. TET tetracycline, DOX doxycycline, CTC chlortetracycline, DOX doxycycline, MIN minocycline, MET metacycline, TIG tigecycline, OMA omadacycline, ERA eravacycline, EFF efflux pump, RPP ribosomal protection protein, DES1 type 1 tetracycline destructase DES2 type 2 tetracycline destructase.

population containing these strains, this dynamic would be predicted to give EFF strains a selective advantage when exposed to first-generation drugs, leading to their expansion relative to the other mechanisms; however, in the context of third-generation drugs, this would be reversed in favor of the DES1 strains because despite DES1 strains having the lowest growth rate, they are the only mechanism that confers resistance to these drugs.

## Barcode sequencing enables accurate measurement of strain relative abundances in mixed cultures

To directly evaluate the population dynamics predicted by our monoculture assays, we next grew our strains in mixed cultures exposed to different tetracycline antibiotics. To do this, we expanded on our previously described library competition sequencing protocol[61], and developed a high-throughput barcode sequencing scheme we call CompAReSeq (Fig. 3a; Supplementary Data 4). This method quantifies the relative abundance of near-identical strains that differ only by the gene of interest and a 7 bp gene-specific DNA barcode. Following the growth of a culture containing multiple strains, plasmid DNA is extracted then the first PCR reaction appends a 6 bp sample-specific barcode and a 19 bp unique molecular identifier downstream of the 7 bp gene-specific barcode. Following PCR clean-up, the second PCR reaction adds sequencing adapters and increases amplicon concentration. The resulting amplicons are then purified, pooled, and sequenced. The relative abundance of each strain in a mix is quantified by dividing the count per gene-specific barcode by the total count of all barcodes in the sample.

To measure the precision and accuracy of this method, we first created three pools of DNA fragments containing the 25 gene-specific barcodes mixed at known amounts, covering a 1000-fold range of relative abundances. We then processed these pools using our

CompAReSeq protocol and compared the measured abundances to the known input abundances. We observed a tight correlation between the input abundance and the abundance inferred by sequencing (Fig. 3b; slope = 0.97, $r^2 = 0.97$, $p$-value = $2 \times 10^{-16}$), validating our method for accurate quantification of strain relative abundances.

## Each generation of tetracycline antibiotics selects a different resistance mechanism

We then applied CompAReSeq to multiplexed competition assays, in which mixtures of our tetracycline-resistant *E. coli* strains were grown together with different tetracycline drugs. First, we prepared a strain mixture composed of all 25 strains (ALL mixture) pooled at equal relative abundances, in triplicate. Each replicate was validated by CompAReSeq to confirm that all strains were at equal relative abundances, and that aliquots were frozen. For the competition experiments, mixture aliquots were inoculated into panels containing tetracycline antibiotics at varying concentrations (0.5x, 1x, or 4x MIC susceptible empty vector control) in triplicate, and grown at 37 °C for 20–48 h. We predicted that antibiotics from each generation would impose different selective pressures on each strain, favoring the growth of some mechanisms over others, resulting in the relative abundances of favored strains to increase and the others to decrease.

As a baseline, we first evaluated the ALL mixture following 20 h of growth in tetracycline-free media (Supplementary Fig. 5a). We observed no significant change in the relative abundance of the summed abundance of the DES1 or RPP mechanisms. Though we observed a significant increase in the DES2 mechanisms and a decrease in the EFF mechanism (pairwise t-test Bonferroni corrected $p < 0.0005$, and $p < 0.005$, respectively), these effects only constituted changes in the relative abundance of approximately ±5% (DES2 = 24.7% to 29.1%; EFF = 23.6% to 18.4%) (Supplementary Fig. 5b). Each mechanism had

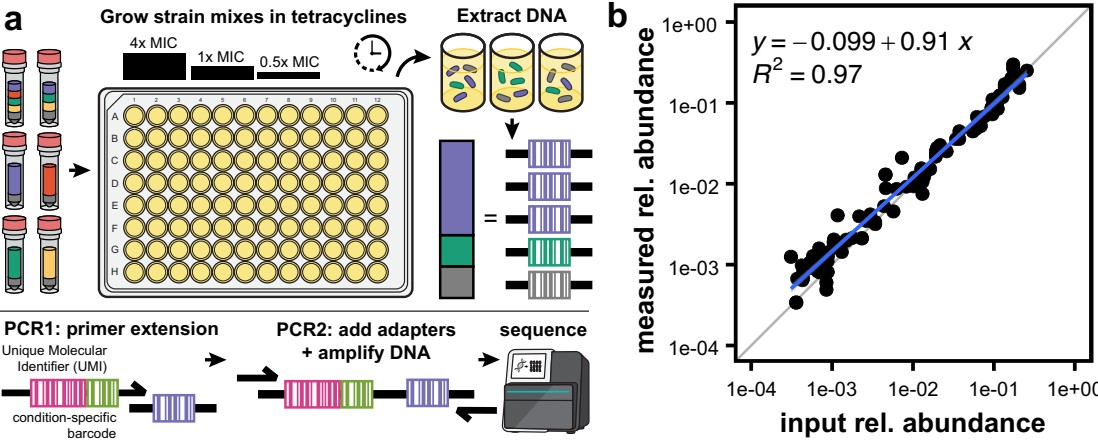

**Fig. 3 | Accurate quantification of barcode relative abundance. a** Schematic of the CompAReSeq barcode sequencing method. Each strain includes a unique 7 bp barcode on the pZE24 plasmid. Strains harboring tetracycline resistance genes, plus an empty plasmid control, were mixed at equal relative abundances. These were then inoculated into media containing antibiotics at different concentrations. Following growth for 20–48 h, plasmid DNA was extracted and prepared for sequencing using two PCR steps that added sample-specific barcodes and UMIs, then added sequencing adapters and amplified DNA. The output amplicons were then sequenced, and the number of UMIs identified per barcode was counted. **b** Relationship between input barcode abundance and measured barcode abundance by sequencing. Each point represents a barcode count, from one of three replicate pools. The blue line indicates the linear trend line and the gray line indicates a slope = 1 (i.e., perfect measurement of input abundance).

individual strains whose relative abundance significantly increased or decreased in relative abundance, but the magnitude of these differences did not exceed ±2% of the strains' starting abundances (Supplementary Fig. 5c). Therefore, we concluded that these changes in the absence of tetracycline antibiotics were not large enough to alter the overall population structure. This suggests that the growth rate differences observed between mechanisms in monoculture are not sufficiently large to influence population structures at the timescales and/or cell densities employed in the mixed-culture competitions.

Next, the ALL mixture was grown in increasing concentrations of tetracycline (first-generation). We observed a direct relationship between the tetracycline concentration and the relative abundance of the EFF strains (Fig. 4a). At 4x MIC, the EFF strains expanded to +240.8% of their starting relative abundance, comprising an average of 80.6% of the total population. This expansion of EFF, however, was not uniform across all EFF strains. The 4x MIC post-competition mixture was primarily composed of *tet*(B) (46.2%), *tet*(E) (21.8%), and *tet*(G) (11.1%), whose relative abundances increased +200–1000% of their starting abundances. The other EFF strains decreased in relative abundances (*tet*(39) = 0.6%, *tet*(A) = 0.5%, *tet*(L) = 0.3%), on par with the decreases for strains from other mechanisms. These results show that tetracycline is preferentially selected for EFF strains over the other tetracycline-resistance mechanisms, despite the other mechanisms conferring similar MICs. This matches the discovery of EFFs in pathogenic isolates shortly after the introduction of first-generation tetracyclines in the clinic[31–33].

When this same ALL mixture was instead grown in increasing concentrations of doxycycline (second-generation) the relative abundances of RPPs increased (Fig. 4b). At 4x MIC, RPPs expanded +227.5% of their starting relative abundance, comprising an average 87.4% of the final mixture. As with tetracycline and EFFs, this expansion was not uniform across RPP strains. The most abundant strains in the 4x MIC post-competition culture were *tet*(W) (49.3%) and *tet*(O) (27.9%), while the relative abundances of the other RPP strains decreased in relative abundance. This trend held when the ALL mixture was grown with minocycline, another second-generation tetracycline (Supplementary Fig. 5d). Again, RPPs increased in relative abundance to constitute a majority of the culture (4x MIC = 93.1%), and the same RPP strains were most abundant (*tet*(W) = 43.9%, *tet*(O) = 40.4%). This preferential selection for RPPs by second-generation tetracyclines suggests that

EFFs are not sufficient to confer resistance to this drug class. This observation matches the intended use of second-generation tetracyclines—to overcome resistance via efflux—and the historical discovery of RPPs in pathogenic isolates shortly after the clinical introduction of second-generation drugs[38,39].

When the ALL mixture was grown in tigecycline (third-generation), we instead observed that DES1 strains increased in relative abundance (Fig. 4c). At 4x MIC, the DES1s expanded +333.7% their starting relative abundance, comprising an average 94.9% of the final mixture. The most abundant strains of the 4x MIC post-competition mixture were *tet*(X12) (56.2%) and *tet*(X3) (37.6%), while the other DES1 strains decreased in relative abundance. This preferential selection for *E. coli* expressing DES1 genes by third-generation tetracyclines matches the discovery of DES1s shortly after the clinical introduction of third-generation drugs[43,44].

## Mechanism-specific strain mixes recapitulate the dynamics observed in the ALL-strains mixture

To further investigate intra-mechanism growth dynamics, we prepared mechanism-specific strain mixtures that contained only strains from a given mechanism, plus the empty vector control. Additionally, to control for any confounding effects from having two tetracycline-inactivating resistance mechanisms in the same population, we also prepared a mixture that contained every strain used in the ALL mix except DES2 strains (-DES2 mix). These mixtures were inoculated in media containing tetracycline (first-generation), doxycycline (second-generation), minocycline (second-generation), or tigecycline (third-generation) at 0.5x, 1x, or 4x the MIC of the empty vector strain, and grown at 37 °C for 20–48 h (Supplementary Fig. 6).

We observed near-identical dynamics in the mechanism-specific mixtures as in the ALL mixture. For example, the dynamics of the EFF-specific mix when grown in tetracycline (Fig. 4d) are highly similar to the EFF dynamics in the ALL mixture (Fig. 4a), including specific changes such as *tet*(G) being the most abundant strain at 1x and 2x MIC but then being the third most abundant at 4x MIC. The dynamics of the RPP-specific mix with doxycycline (Fig. 4e) and the DES1-specific mix with tigecycline (Fig. 4f) were also highly similar to those of the ALL mixture, as was the -DES2 mix with all drugs (Supplementary Fig. 7). Together, these results show that the same strain dynamics that occur in the ALL mixture also occur in the mechanism-specific mixtures. This

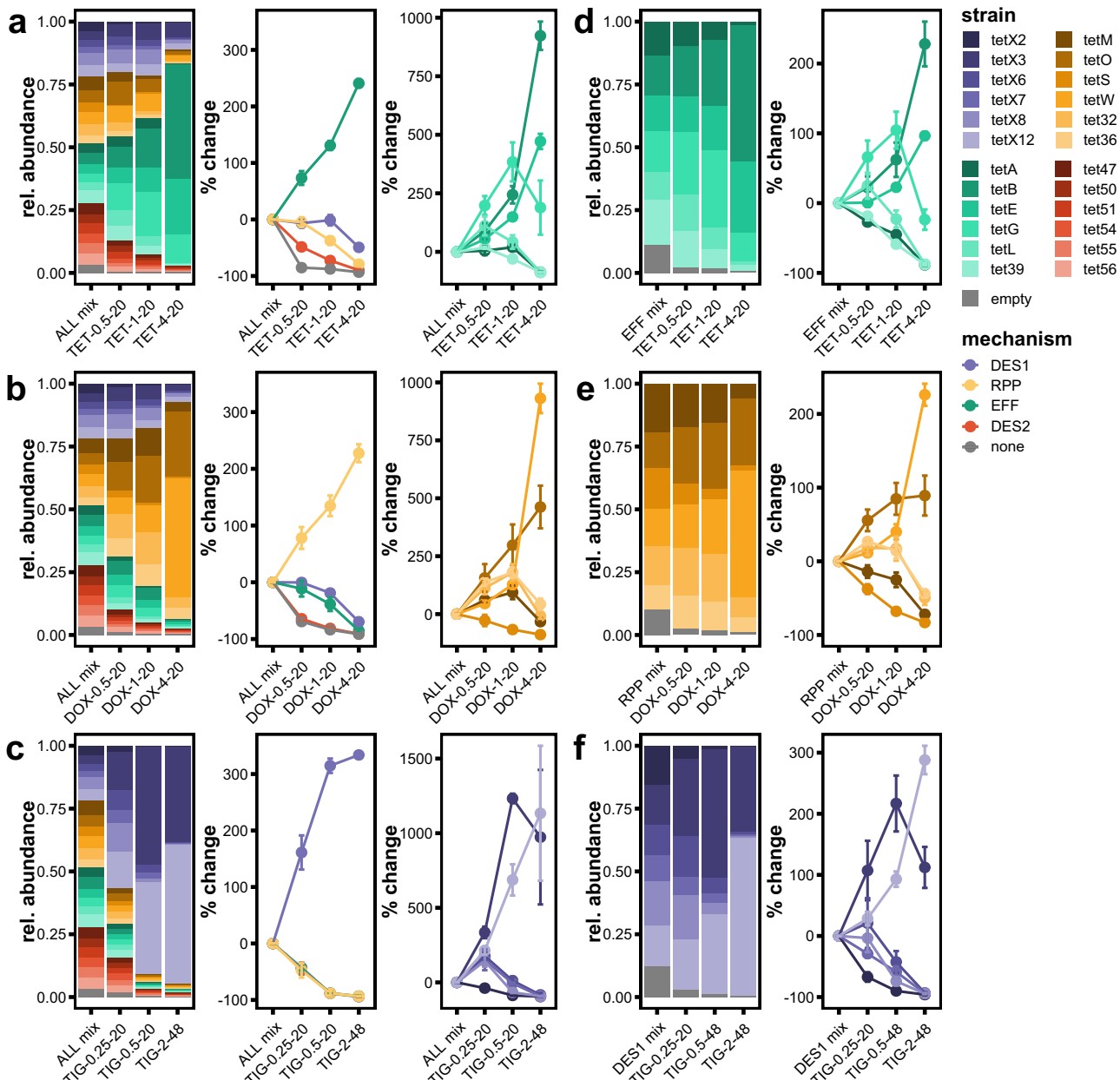

**Fig. 4 | Each generation of tetracycline antibiotics selects for different resistance mechanisms from the same starting mixture.** Comparisons of strain relative abundance between the starting strain mixture and the post-competition culture after 20–48 h growth at 0.5x, 1x, and 4x empty strain MIC for each antibiotic. Results for the mix containing all 25 strains (ALL mix) grown with **a** tetracycline (1st gen), **b** doxycycline (2nd gen), and **c** tigecycline (3rd gen). Results for the mechanism-specific mixes **d** EFF with tetracycline, **e** RPP with doxycycline, and **f** DES1 with tigecycline. X-axis labels denote antibiotic-concentration-timepoint (hours). Left panel: barplots of the relative abundance of each strain in the starting mix, then after growth in the antibiotic selective conditions at 0.5x, 1x, and 4x the MIC of the empty vector strain. Bar height represents the average of three replicates. Middle panel: the percent change in the summed relative abundances of each strain belonging to a given mechanism relative to the starting mixture. Right panel: the percent change in the relative abundance of specific strains from a given mechanism. Points represent the average of three replicates, and error bars represent the standard deviation. TET tetracycline, DOX doxycycline, TIG tigecycline, EFF efflux pump, RPP ribosomal protection protein, DES1 type 1 tetracycline destructase, DES2 type 2 tetracycline destructase.

suggests that the preferential selection of different mechanisms in the ALL mixture is not due to direct interactions between mechanisms or their byproducts. Instead, the selection for a mechanism within the ALL mixture is due to the suppression of strains from other mechanisms, evidenced by the strains of the most advantageous mechanism growing as if there were no other mechanisms there.

**Strains are selected in a mechanism-first, MIC-second manner**
We next evaluated features that might contribute to the selection of specific *E. coli* strains by each antibiotic. In particular, we were

interested in differences between strains within the same mechanism which could explain why some increase in abundance while others decrease. In the ALL mix tetracycline competition, the EFF strains with the highest relative abundances at 4x MIC−*tet*(B) (46.2%) and *tet*(E) (21.8%)−also had the highest EFF individual strain tetracycline MICs at 256 µg/mL (Supplementary Fig. 8). These were followed by *tet*(G) (11.1%) with a tetracycline MIC of 64 µg/mL, and the remaining EFF strains with relative abundances <1% had tetracycline MICs of 32 µg/mL. This suggests that MIC is predictive relative abundance (i.e., higher MIC = greater relative population abundance). However, the

DES1s *tet*(X3) and *tet*(X12) also had tetracycline MICs of 256 µg/mL, but their relative abundances (5.8% and 2.3%, respectively) were far lower than those of the EFFs *tet*(B) and *tet*(E).

Similarly, during doxycycline selection, the RPP strain with the highest relative abundance was also the RPP strain with the highest doxycycline MIC, *tet*(W) (49.3%) at 64 µg/mL. This was followed by *tet*(O) (27.9%) at 32 µg/mL, and the RPP strain with the lowest doxycycline MIC had the lowest relative abundance, *tet*(S) (0.4%) at 8 µg/mL. However, the DES1 strain *tet*(X12) had a doxycycline MIC of 128 µg/mL−greater than that of *tet*(W)−and yet had a relative abundance of only 1.7%. This pattern was also observed with minocycline. Lastly, the DES1 strains with the highest relative abundances also had the highest DES1 tigecycline MICs (*tet*(X3) = 47.0%, 8 µg/mL; *tet*(X12) = 36.4%, 8 µg/mL), and the remaining DES1 strains with MICs less than or equal to 4 µg/mL had relative abundances <5%. Taken together, these results show that MIC alone is not predictive of relative abundance after tetracycline selection. However, when comparisons are first restricted to the mechanism preferentially selected for by the antibiotic generation (e.g., EFF for tetracycline selection), then MIC can be used to predict which strains will have the highest relative abundances.

### Sequential transfer of competition cultures to different tetracyclines reveals DES1 strains retain overrepresentation

Having established each tetracycline generation selects for specific resistance mechanisms, we investigated what dynamics occur when a population first grown with one drug is then transferred to media containing a different generation of tetracycline. To do this, we implemented a transfer experiment where the output of one competition was used as the input mixture for competition in a different antibiotic. The ALL mixture was first grown in 1x MIC of tetracycline, doxycycline, or tigecycline for 24 h, then an aliquot of the culture was diluted 1:100 into new media supplemented with a different antibiotic. This process was repeated a second time with another antibiotic, such that the mixture was subjected to every combination of tetracycline, doxycycline, and tigecycline transfer (Fig. 5).

Transfer from tetracycline to doxycycline to tigecycline recapitulates the results observed in the single-antibiotic competition, where EFFs increase in relative abundance in tetracycline, then RPPs in doxycycline, then DES1s in tigecycline (Fig. 5a). However, when the order of transfer was instead doxycycline to tetracycline then tigecycline, the relative abundance of RPPs increased in doxycycline as expected, but then continued to expand in tetracycline (Fig. 5c). The most lasting advantage observed was for third-generation tigecycline and DES1s. Whenever a mixture was grown in tigecycline, regardless of what antibiotics it had been grown in previously, the DES1 strains expanded to represent the majority of the culture. Further, when the DES1-majority culture was transferred to an earlier-generation drug, DES1 strains still maintained that high relative abundance (Fig. 5b, d–f). We conclude that because DES1 strains confer resistance to all tetracycline antibiotics, while the EFF and RPP strains do not with tigecycline, the selective advantage of EFFs and RPPs in earlier-generation drugs is not sufficiently large enough to overcome the effects of DES1's head start with third-generation drugs; however, this may be confounded by small carryover of tigecycline with the cells into subsequent competitions.

### Bacterial genomes encode multiple tetracycline-resistance mechanisms

The results of our mixed-culture competition experiments suggest that tetracycline-resistance mechanisms are not functionally redundant, but advantageous with specific tetracycline generations. This suggests that bacteria may acquire and retain more than one genetically encoded tetracycline-resistance mechanism. To assess the co-occurrence of tetracycline-resistance mechanisms, we collected 479 bacterial genome assemblies from the NCBI genome database that

contain DES1 genes (Supplementary Data 5) and searched these for EFF and/or RPP genes. We observed that 45% (*N* = 218) of genomes contain genes for more than one tetracycline-resistance mechanism (Supplementary Fig. 9a, b). Among co-occurring mechanisms, the combination of DES1 + RPP was the most frequent (*N* = 188), followed by DES1 + EFF (*N* = 26). For example, the DES1 *tet*(X2) and RPP *tet*(Q) were encoded on the same contig in *Bacteroides fragilis* and *Parabacteroides distasonis*, and the DES1 *tet*(X5) and EFF *tet*(Y) were encoded on the same contig in *Acinetobacter indicus* (Supplementary Fig. 9c). We observed all three mechanisms, DES1 + EFF + RPP, encoded in genomes of *Acinetobacter*, *Bacteroides*, and *Escherichia* species (*N* = 4). An *Acinetobacter variabilis* genome encoded the DES1s *tet*(X3) and *tet*(X5), the RPP *tet*(M), and the EFF *tet*(Y). Together, this shows that DES1-encoding bacteria frequently encode multiple tetracycline-resistance mechanisms, supporting the argument that it may be beneficial to acquire or retain different mechanisms as they are specialized for different tetracycline generations.

## Discussion

Using our high-throughput barcode sequencing method, CompAReSeq, we quantified the changes in the relative abundances of isogenic *E. coli* strains encoding different tetracycline-resistance mechanisms in mixed cultures. We demonstrated that each tetracycline generation preferentially selects for *E. coli* expressing different resistance mechanisms: tetracycline (first-generation) selects for EFFs, doxycycline and minocycline (second-generation) select for RPPs, and tigecycline (third-generation) selects for DES1s. This is consistent with the history of the clinical tetracycline resistome[30], as well as the development strategies for each generation of antibiotic to counter the mechanism(s) conferring resistance to the previous generation[21]. Therefore, we conclude that the emergence of each mechanism in human pathogens was due to specific dynamics between tetracycline drug molecules and resistance mechanisms. This effect was confirmed to be primarily driven by mechanism and antibiotic structure, not by MIC or growth rate defects alone.

Our results are consistent with prior studies on how later-generation tetracyclines alter environmental resistomes, on the epidemiology of tetracycline resistance in pathogens, and on the relative fitness of genes belonging to different resistance mechanisms. For example, Tang et al. reported that early-generation tetracyclines stimulate the production and proliferation of antibiotic resistance genes in soil environments to a greater extent than later-generation tetracyclines, an effect driven by the D-ring modifications found in third-generation tetracyclines[62]. This provides a structural explanation for the dominant selection of DES1 during exposure to tigecycline and sustained high abundance during subsequent exposures to doxycycline and tetracycline. In addition, Da Cunha et al. showed that the emergence of *Streptococcus agalactiae* human infections during the 1960s is linked to the expansion of a few clones that encoded the RPP *tet*(M) gene[63]. This is consistent with our hypothesis that the deployment of second-generation tetracyclines selected for pathogens encoding RPPs results in their expansion from within diverse bacterial populations. Our results are also consistent with their findings that both *tet*(X) and the RPP *tet*(O) impose a cost on *E. coli* by lowering growth rate, but that *tet*(X) poses a greater apparent burden than *tet*(O).

The differences in growth rate and substrate specificities between *E. coli* strains expressing DES1 or DES2, despite their similarities in mechanism and structure, could be due to DES2 enzymes having an additional C-terminal gatekeeper helix that permits greater selectivity of substrates[46]. The DES1 enzymes, which lack this gatekeeper helix, may oxidize additional substrates other than tetracyclines, resulting in the increased membrane permeability and decreased fitness shown by Chen et al.[64]. Palmer et al. found that photoinactivation of tetracyclines can reverse selection outcomes of tetracycline-resistant and

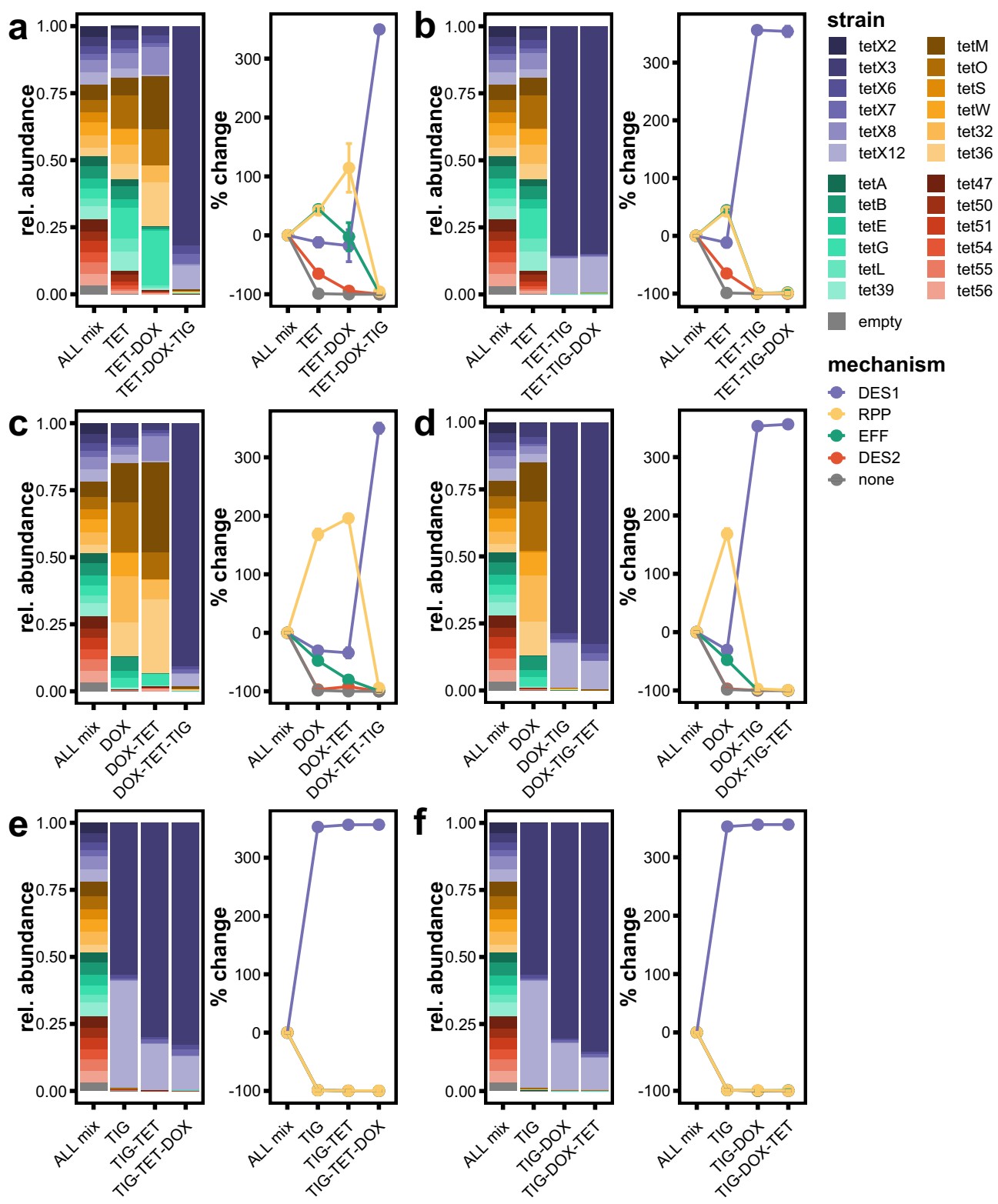

susceptible strains of *E. coli*[8] suggesting that tetracycline chemical conversion or breakdown products will result in more complex dynamics determining microbial population dynamics[65–67]. Future work into potential off-target oxidation and the influence of tetracycline inactivation products produced by DES1 enzymes is warranted.

When recombinant proteins are heterologously expressed, gene sequences are frequently optimized for overexpression by substituting synonymous codons for those that occur more frequently in the host system. We chose not to artificially codon optimize our tetracycline-resistance genes for *E. coli* for three reasons: first, previous reports on these genes used the wild-type sequences, so we continued

**Fig. 5 | DES1 strains continue to dominate the population when transferred into earlier-generation tetracyclines.** Comparisons of strain relative abundance during competition transfer experiments. The mix containing all 25 strains (ALL mix) was first grown with 1x MIC tetracycline, doxycycline, or tigecycline for 24 h. Then 2 μL of the output of these competitions were inoculated into media containing a different antibiotic at 1x MIC and grown for another 24 h. The remainder was processed by CompAReSeq to determine strain relative abundance. This was repeated a third time, to generate results for all possible combinations of transfers. X-axis labels denote the order of drugs transferred. Results ALL mix grown starting with **a**, **b** tetracycline, **c**, **d** doxycycline, and **e**, **f** tigecycline. x-axis labels denote the order of drugs. Left panel: barplots of the relative abundance of each strain in the starting mix, then after growth in the antibiotic selective conditions. Bar height represents the average of three replicates. Right panel: the percent change in the summed relative abundances of each strain belonging to a given mechanism relative to the starting mixture. Points represent the average of three replicates, and error bars represent the standard deviation. TET tetracycline, DOX doxycycline, TIG tigecycline, EFF efflux pump, RPP ribosomal protection protein, DES1 type 1 tetracycline destructase, DES2 type 2 tetracycline destructase.

to use those same sequences for consistency; second, by using wild-type sequences we can infer that these results are similar to what would be observed in nature following the horizontal transfer of these genes into the same *E. coli* host; third, many tetracycline-resistance genes are found in a wide range of host backgrounds including both Gram-negative and Gram-positive taxa (Supplementary Fig. 1c, d)[60,68,69]. Nevertheless, differences in codon usage of expressed constructs could be an important variable to consider in future evaluations of fitness benefits.

Our analyses were simplified to a single host organism, *E. coli* DH5αZ1, differing only by the resistance genes and antibiotics used. In natural populations gene effects can vary in different genetic backgrounds and environments[12,65,70,71]. Previous reports have shown that there are different costs associated with different species expressing the same DES1 gene[72], or the same species expressing the same gene in different plasmids[73]. For example, Jiang et al. inserted plasmid constructs encoding *tet*(X6) into *Escherichia coli*, *Salmonella* Enteritidis, and *Proteus mirabilis*, and found that while the MICs of *E. coli* and *S.* Enteritidis increased to a greater extent than *P. mirabilis* (both 8-fold vs. 2-fold), they suffered greater costs including decreased growth rate, biofilm formation, and virulence[72]. In addition, it is possible that our strains had differential fitness under variable conditions not tested, such as temperature and pH[74,75]. Lastly, although the genes we used are a representative subset of the overall sequence and taxonomic diversity of the tetracycline resistome, we do not exhaustively cover all known resistance genes and sequence variants. Further work is required to determine whether the trends observed in this study hold true for different plasmids and hosts, to consider additional phenotypes that contribute to fitness, and to evaluate additional tetracycline-resistance genes. Nevertheless, our CompAReSeq methodology provides the framework for making these additional comparisons.

An additional limitation to inserting all our genes into the same plasmid system is that our experiments may not capture potential regulatory dynamics that influence the timing and magnitude of gene expression. For example, the *tet*(A) efflux pump has been observed under the control of the Tet repressor (TetR), such that expression of the pump is induced by the binding of anhydrotetracycline[76,77]. In tetracycline biosynthesis, anhydrotetracycline is a precursor to the final tetracycline molecule, which makes this regulatory framework advantageous for tetracycline-producing bacteria as it ensures that newly synthesized tetracycline is immediately pumped out of the cell; however, it can be disadvantageous because anhydrotetracycline is also a natural decay product of tetracycline. It has been shown that needless activation of *tet*(A) expression by tetracycline that has decayed to anhydrotetracycline incurs growth defects that put these strains at a competitive disadvantage relative to susceptible strains[78]. Further complicating matters, anhydrotetracycline is a competitive inhibitor of DES1 and DES2 enzymes, a fact which we have leveraged to develop anhydrotetracycline derivatives as an adjuvant in potential combination therapy with tetracyclines[79,80]. These multi-faceted dynamics of regulation, degradation, and inhibition are likely to result in complex and interesting population dynamics, which can be directly investigated in the future using our CompAReSeq method.

While acknowledging that the complexities of selective pressures and microbial growth dynamics will be much more complicated in the real world, we nevertheless stress the importance of using isogenic strains to minimize variables and ensure fair comparisons. Our experimental design captures unique aspects of evolution during selection for specific functions. Fitness is related to an organism's current environment[1,2], and several experimental evolution setups have shown that antibiotic-susceptible bacteria outcompete more resistant strains when the environment does not require those functions for survival[7–9]. Here, we focused on two important aspects of fitness in the context of antibiotic perturbation: growth rate and susceptibility. *E. coli* strains encoding EFFs are as susceptible to third-generation drugs as the empty vector control; however, this disadvantage for EFFs, along with the advantage of DES1s in these drugs, is irrelevant in the context of first-generation drugs to which both confer resistance. Instead, EFFs were predicted to be more fit than the other resistance mechanisms in the context of first-generation drugs, as shown in our competition experiments with tetracycline. However, during our transfer competition experiments—which represent a population affected by changing environmental conditions—DES1 strains remained at the highest abundances for at least 24 h following transfer into tetracycline and doxycycline post-tigecycline selection. This evidence indicates a lag period, where the substantial advantage that DES1s provide versus EFFs and RPPs under tigecycline selection enables a period of maintenance of strains encoding DES1s even when switched to conditions where they are otherwise relatively less fit. This is likely because EFFs and RPPs do not have as large a relative advantage versus DES1s in the presence of tetracycline and doxycycline, respectively. However, we predict the advantages of EFFs and RPPs versus DES1s would eventually lead to their relative abundances increasing over a longer timespan than investigated here (24 h), and further investigation into the dynamics at play during this transitionary phase is warranted.

A similar evolutionary arms race has been extensively described between the deployment of new generations of beta-lactam antibiotics and the discovery of beta-lactamase antibiotic-inactivating enzymes[81–83]. Concurrent with the historical parallels, many of the evolutionary dynamics we have described for the tetracycline resistome also parallel the beta-lactamases. For example, the emergence of new beta-lactamases has been linked to expanded substrate specificities[84], the limited dissemination of some carbapenemases has been linked to their greater costs[85,86], and single isolates frequently encode multiple beta-lactamases[87]. Another feature of beta-lactamases that has been extensively studied is the impact of individual amino acid substitutions on substrate specificity[82,88–92]. We have previously shown that just a few substitutions can also have a large impact on the substrate specificity of TDase enzymes, with single alanine substitutions resulting in the loss of tigecycline activity in Tet(X7)[56]. Additionally, directed evolution experiments have shown that resistance genes from all tetracycline-resistance mechanisms can mutate to acquire at least low-level resistance to tigecycline, but only Tet(X) can do so without compromising its activity against earlier-generation drugs[93]. These resistomes differ in many important ways (e.g., these dynamics occurred in tetracyclines for multiple mechanisms while for beta-lactams it was only enzymatic inactivation, and beta-lactamases are frequently secreted into

environments while DES1s and DES2s likely are not) and other factors can impact the spread of resistance genes[94]. Nevertheless, we believe that the parallels are numerous and important enough to evidence larger trends in the evolution of antibiotic resistance.

Previous experimental methods for comparing susceptible and resistant bacterial strains in mixed cultures differentiated between strains by additional antibiotic selective or fluorescence markers[12,65]. However, these approaches are limited by low throughput, an inability to discriminate between more than a few strains at a time, and the need for substantial phenotypic differences between the strains in addition to the genes of interest. High-throughput methods for studying the evolution of resistance in bacterial populations are increasingly common, but these methods have typically examined strains that differ by of specific mutations acquired during adaptation to antibiotic conditions[12,65]. To our knowledge, this is the first report directly comparing different mechanisms and genes involved in resistance to the same drug class in mixed-culture competition experiments. Our high-throughput barcode sequencing method represents a new approach for this kind of analysis and has the advantage of being able to be multiplexed to 96-well plates for the competition cultures and downstream PCR reactions. We successfully demonstrated the use of 25 isogenic *E. coli* strains, and this approach could be expanded to thousands of strains requiring that they only differ by short (here, 7-bp), unique DNA barcodes. Owing to this flexibility, CompAReSeq provides a robust platform for comparing resistance genes to those of other antibiotic classes (e.g., beta-lactamases), or genes encoding other readily selectable functions, and could be useful for clinical analysis and biosynthetic engineering.

## Methods
### Plasmids, strains, media
Tetracycline-resistance genes were cloned into the *KpnI* and *MluI* sites of the pZE24 plasmid (Expressys). The plasmid's $P_{lac/ara-1}$ promoter is regulatable using IPTG and arabinose in the *E. coli* DH5αZ1 background[95] (Supplementary Fig. 1e) and maintained using kanamycin. For high expression from the $P_{lac/ara-1}$ promoter, 1 mM IPTG fully relieves repression by LacI[95]. Assembly was performed using the HiFi DNA Assembly kit (NEB) as per the manufacturer's instructions. Gene-specific barcodes (Supplementary Data 4) and flanking primer-binding regions (TAGCCATGTCATAGACGTCC-XXXXXXX-GTACGTTGATCAAG TCCCGA, where XXXXXXX denotes the 7-bp gene-specific barcode) were inserted at the *SpeI* site using the Q5 Site-Directed Mutagenesis kit (New England Biolabs) with mutagenic primers designed using NEBaseChanger v1.3.2 (Supplementary Data 6). Each barcode had a GC content between 40% and 60% and a minimum Hamming distance of 2. Plasmid constructs were transformed into chemically competent *E. coli* DH5αZ1 (Expressys) by heat shock, then streaked onto cation-adjusted Mueller Hinton II (CAMH) agar (BBL; Media format: Beef extract 3 g/L, acid hydrolysate of casein 17.5 g/L, starch 1.5 g/L) supplemented with 50 μg/mL kanamycin (kan). Individual colonies were then prepared into 25% glycerol freezer stocks and stored at -80 °C. Correct assembly of the tetracycline-resistance gene and DNA barcode was verified by Sanger sequencing (GeneWiz) or whole-plasmid sequencing (Plasmidsaurus).

When needed for experiments, freezer stocks were streaked onto CAMH+kan50 agar, then 2–3 colonies were inoculated into 5 mL CAMH +kan50 broth, and grown overnight at 37 °C with shaking. Overnight cultures were diluted into CAMH+kan50 broth supplemented with 1 mM IPTG, then grown until exponential phase ($OD_{600}$ = 0.3–0.8). For high expression from the $P_{lac/ara-1}$ promoter, 1 mM IPTG fully relieves repression by LacI[95]. These were then diluted to a standard concentration prior to inoculation in test conditions.

**Genera comparison across mechanisms.** Protein sequences from each resistant gene with ≥99% sequence identity were retrieved from the NCBI database using Blastp and Biopython[96]. The genera associated with each sequence were annotated using custom Python scripts in conjunction with the ETE3 toolkit[97]. To determine the Gram classification of each bacterial genus, entries were manually cross-referenced with the BacDive database[98]. As DES2 genes were identified with functional metagenomics, and only sequence has been associated with a genus (*Legionella*), this mechanism was excluded from the visualization.

### Antibiotic susceptibility testing (AST)
ASTs were performed with the microbroth dilution method, as per CLSI guidelines[99]. MIC panels were prepared in 96-well flat-bottom microplates (Corning) by twofold serial dilution of tetracycline antibiotics in CAMH+kan50 broth, then stored at −80 °C for <6 months. On the day of the experiment, MIC panels were thawed at room temperature, and overnight cultures were grown in fresh CAMH+kan50 broth supplemented with 1 mM IPTG to exponential phase ($OD_{600}$ = 0.3–0.5, ~2.5 h). Cultures were then diluted to a standard concentration and inoculated into each panel at a 1:1 ratio, such that each well had a final concentration of: 50 μg/mL kanamycin, 1 mM IPTG, ~5 × 10⁵ CFU/mL cells (0.5 MacFarland), and variable concentrations of the antibiotic of interest. Each test was performed in triplicate, with no-antibiotic and no-cell control wells. Inoculated panels were incubated at 37 °C for 20 h, then scored by eye and using a Synergy H1 plate reader (BioTek).

### Measuring growth rate
Overnight cultures were grown in fresh CAMH+kan50 broth supplemented with 1 mM IPTG to exponential phase ($OD_{600}$ = 0.3–0.5, ~2.5 h), then diluted to $OD_{600}$ = 0.1, and inoculated into each 96-well panel at a 1:1 ratio. To avoid edge effects only the interior wells included cells, and the exterior wells were filled with cell-free broth. Plates were sealed with Breathe-Easy membranes (Sigma-Aldrich) and then incubated at 37 °C with continuous shaking and $OD_{600}$ measurements taken every 5 min for 20 h using a Synergy H1 plate reader (BioTek). The maximal growth rate was calculated from this plate reader data using GrowthRateR (https://github.com/kevinsblake/GrowthRateR). This function log-transforms growth curves and generates a rolling regression with a shifting window of 1 h, such that the maximum slope of any of the regressions is the maximal growth rate. Growth rates were compared using Prism[100] with an ordinary one-way ANOVA with the FDR method of Benjamini and Hochberg.

### Generating DNA fragments to validate CompAReSeq barcode sequencing method
Triplicates of each of the 25 plasmid constructs were grown overnight in CAMH+kan, and plasmid DNA was extracted using a QIAquick® Spin Miniprep Kit (Qiagen), resulting in three independent extracts of each construct. The regions containing the gene-specific barcodes were PCR amplified for 35 cycles using primers TGAAGCCAGTTACCTTCGG and CTGCTTGCCGAATATCATGG and Q5 Hotstart DNA Polymerase 2x Master Mix (NEB), as per manufacturer's instructions. PCR products were separated on a 2% agarose gel, and the band corresponding to the expected amplicon size was excised then DNA extracted using the QIAquick Gel Extraction Kit (Qiagen). The concentration of the resulting amplicons was quantified using the QuBit kit (ThermoFisher). These were then pooled at known amounts to generate barcode mixtures of known compositions, which then underwent barcode sequencing.

### CompAReSeq barcode sequencing
The first PCR (PCR1) used 1 μL of extracted plasmid DNA product and added sample-specific barcodes and unique molecular identifiers (UMI) using the primer GGCAAATAAAACGAAAGGCTCA-NNNNN NNNNNNNNNNNNN-XXXXXX-TCGGGACTTGATCAACGTAC, where

each N refers to a base equally likely to be A, C, T, or G, forming a UMI, and XXXXXX denotes the 6 bp sample-specific barcode (Supplementary Data 4). PCR1 was performed using Q5 HiFi DNA Polymerase as per manufacturer's instruction, with a melting temperature of 65 °C, an extension time of 15 s, and 3 cycles. PCR1 products were then purified using Agencourt Ampure XP beads (Beckman Coulter). The second PCR (PCR2) used 5 μL of this purified PCR product to amplify DNA and add adapters for sequencing, using primers AATGATACGGCGACC ACCGAGATCTACAC-i5-ACACTCTTTCCCTACACGACGCTCTTCCGAT CT-TAGCCATGTCATAGACGTCC and CAAGCAGAAGACGGCATACGA GATG-i7-TGACTGGAGTTCAGACGTGTGCTCTTCCGATCT-GGCAAATA AAACGAAAGGCTCA, where i5 and i7 indicate Illumina sequencing adapters. (Note: phasing was not used in these experiments but can be added.) PCR2 products were separated on a 3% agarose gel, and the band corresponding to the expected amplicon size was excised. Amplicons were purified using the MinElute Gel Extraction Kit (Qiagen), and DNA concentration was quantified using a Qubit (Thermofisher). Extracted DNA amplicons were then pooled and sequenced using an Illumina Miniseq to a depth of at least 50,000 $2 \times 150$ paired-end reads per sample. (Note: the protocol can be adapted for single-end sequencing, or other sequencing methods, by modifying the adapters used for the PCR2 primers.) Sequencing reads were demultiplexed by barcode, then had adapters removed with Trimmomatic v0.38 with the following parameters: *ILLUMINACLIP:NexteraPE-PE.fa:2:30:10:1:true SLIDINGWINDOW:4:20 LEADING:10 TRAILING:10 MINLEN:60*. Clean reads were then processed with the *barseq.py* script, which identifies and counts gene- and sample-specific barcodes and UMIs. The relative abundance of each strain in a mix was quantified by dividing the count per barcode by the total count of all barcodes in the sample. Strains with a relative abundance <0.1% were masked from downstream analyses.

### Generating strain mixtures
Single colonies of each *E. coli* strain were inoculated into 1 mL CAMH +kan50 in 96-well deep-well plates in triplicate and grown overnight at 37 °C with shaking. Overnight cultures were diluted 1:2, $OD_{600}$ was measured using a Synergy H1 plate reader (BioTek), and then cultures were diluted to $OD_{600} = 0.1$ in CAMH+kan50. Strain mixtures were prepared by pooling these diluted cultures at equal volumes (Supp. Data 7), followed by an equal volume of 30% v/v glycerol. These were prepared as 0.5-1 mL aliquots and then frozen at −80 °C until future use.

### Mixed-culture competition experiments
Strain mixture triplicates were thawed on ice, and then 100 μL of each strain mixture was inoculated 1:1 into competition panels, resulting in a final concentration of CAMH+kan50 + 1 mM IPTG + the antibiotic of interest. Inoculated panels were then incubated at 37 °C with shaking at 225 RPM for 20 h. Following incubation, $OD_{600}$ was quantified using a plate reader. If a condition did not reach an $OD_{600} > 0.4$ by 20 h, the test was repeated and instead incubated for 48 h. Plasmid DNA was extracted using the QIAquick® Spin Miniprep Kit or the QIAprep® 96 Turbo Miniprep Kit (Qiagen), then processed by barcode sequencing.

### Transfer competition experiments
Strain mixture triplicates were thawed on ice, each strain mixture was diluted 1:10. then, 100 μL diluted mixture was inoculated 1:1 into competition panels, resulting in a final concentration of CAMH +kan50 + 1 mM IPTG + 1x MIC the antibiotic of interest. Inoculated panels were then incubated at 37 °C with shaking at 225 RPM for 24 h. Following incubation, 2 μL of the resulting culture was inoculated into 200 μL broth containing a different antibiotic at 1x MIC, then grown for another 24 h. The remaining culture was subject to plasmid DNA extraction using the QIAquick® Spin Miniprep Kit or the QIAprep® 96 Turbo Miniprep Kit (Qiagen). This process was repeated to generate each possible combination of transfer between tetracycline, doxycycline, and tigecycline.

### Analysis and visualization of co-occurring tetracycline-resistant mechanisms
479 genome assemblies containing DES1 genes were retrieved from the NCBI genome database using Biopython[96] and custom scripts. Antibiotic resistance genes were annotated with AMRfinder v3.10.42[101], and insertion sequences with Prokka v1.14.6[102]. For co-occurring gene visualization, contigs containing tetracycline-resistance genes were extracted with customized scripts, and the gene plots were generated using gggenome (https://github.com/thackl/gggenomes).

### Data analyses
Hierarchical clustering and visualization of MIC values was done using *pheatmap* v1.0.12 (https://CRAN.R-project.org/package=pheatmap). Figure color palettes generated using *NatParksPalettes* v0.2 (https://cran.r-project.org/web/packages/NatParksPalettes/index.html).

### Statistics and reproducibility
AST and growth rate experiments were performed in triplicate. Mixed-culture competition experiments were performed using three independent strain mixes.

### Reporting summary
Further information on research design is available in the Nature Portfolio Reporting Summary linked to this article.

## Data availability
Source data are provided in this paper.

## Code availability
The data and analysis scripts are available at: https://github.com/dantaslab/2024_Blake_TET-resistome. This repository is also available at: https://doi.org/10.5281/zenodo.14536831[103].

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

## Acknowledgements

This work is supported in part by the National Institute of Allergy and Infectious Diseases (NIAID) of the National Institutes of Health (NIH) through grant 2U01AI123394 awarded to G.D. and T.A.W. at Washington University in St. Louis. N.H.T. is supported by the Intramural Research Program of the Division of Intramural Research, NIAID, NIH. K.S.B. is supported by the National Institute of Diabetes and Digestive and Kidney Diseases (T32-DK007130; PI: N. Davidson). The content is solely the responsibility of the authors and does not necessarily represent the official views of the funding agencies. This work is also supported by K99AI175674 [PI: S.R.S.F.] and T32DK077653 [PI: P. Tarr] to S.R.S.F. We thank the staff at The Edison Family Center for Genome Sciences & Systems Biology at Washington University School of Medicine in St. Louis, including Eric Martin and Brian Koebbe for computational support, and Jacky Theodore, Bonnie Dee, James Anderson, and Keith Page for administrative support. Special thanks to Jessica Hoisington-López and MariaLynn Crosby for managing the high-throughput sequencing core and their assistance in designing the barcode sequencing scheme. Finally, we would like to thank the members of the Dantas lab for helpful general discussions and comments on the manuscript.

## Author contributions

G.D., T.A.W., N.H.T., K.S.B., Y.P.X., V.G., and S.R.S.F. conceived the study design, experiments, and analysis. K.S.B. generated the *E. coli* strains, developed the barcode sequencing method, performed the mono- and mixed-culture assays, analyzed the data, and interpreted the results. Y.P.X. performed the gene co-localization analysis. K.S.B. drafted the article, with critical revisions from G.D., T.A.W., and N.H.T. All authors reviewed and approved the final manuscript.

## Competing interests

The authors declare no competing interests.
