## [Transparent Peer Review file · Nature Communications]

The tetracycline resistome is shaped by selection for specific resistance mechanisms by each antibiotic generation

Corresponding Author: Dr Kevin Blake

Version 0:

Reviewer comments:

Reviewer #1

(Remarks to the Author)

The manuscript by Blake et al. first describes how the selection of tetracycline resistance genes occurs through the selection of specific resistance mechanisms by tetracycline antibiotics of different generations. This resembles the introduction of such antibiotics in the therapeutical arsenal and the emergence of different genes/resistance mechanisms over time. The work may be relevant to understanding some aspects related to the ecology (distribution, redundancy) and evolution (exaptation, phenotypic evolution) of antibiotic resistance.

The predominance of some resistance mechanisms, such as tetracycline efflux pumps, is observed in resistomes of pristine environments, but the rationale beyond these findings has not been addressed. In addition to the novelty of the results, the experimental approach is highly innovative. The high-throughput barcode sequencing protocol (CompAReSeq: COMPetitive Antibiotic Resistance SEQuencing) that can discriminate and quantify genotypes in mixed cultures provides a valuable approach to testing the evolutionary pathways of multiple genes in microbial populations. Although the idea is not new, it is first applied to antibiotic resistance.

The work is interesting, but the interpretation of some results should be revised. Some aspects of the experimental approach should be further detailed.

Some comments for the author's consideration appear below.

The tetracycline resistome. The number of known EFF and RPP genes is higher than that evaluated in this work (EFF: 30 vs. 6; RPP: 12 vs 6), while DS1 are allele variants of tet(X) (Pe.g., PMID: 27065405). The criteria for selecting the pool of tet genes tested are not provided. Note that tet genes (the sample analyzed here and the known pool) are neither equally prevalent nor have the same genetic context preference. While some genes are common in G+ve and G-ve taxa, others are associated with particular phylogenetic backgrounds, which could influence their prevalence and expression and, thus, some experimental results (see below).

Clonal background. Using a single clonal background is a limitation of the work (also recognized by the authors). The burden of the experimental work is impressive, and the possibility of extending the analysis to other strains, either the same or different species, seems unfeasible. However, the reader misses a more profound discussion regarding particular genes. EFF genes analyzed include some exclusively detected in G-ves (A, B, E, and G) and others widely distributed in G+ves (L, 39). The last ones were not predominantly selected in the experiments.

Conversely, RPP genes are common in G+ve and G-ve or confined to G+ve taxa. Your results show a selection bias towards those with a broader distribution. A MIC assay of these constructions in other clonal backgrounds will be welcome to discard weird discussions about MICs. This value could change depending on the taxa background.

Epidemiological data could enrich the discussion about the parallelism between introducing each antibiotic class in the therapeutic arsenal and the emergence of each resistance mechanism. The temporal sequence applies to G-ve and G+ves bacterial species where tetracycline resistance is extremely common (PMID: 25088811).

Parallelism between the selection of tetracyclines and beta-lactams. The authors compare the sequential selection of tetracycline-resistant genes (which are associated with EFF, RPP, and enzymatic inactivation) with some beta-lactams (all linked to enzymatic inactivation and restricted to class A and class B beta-lactamases). This is a false equivalence because of different mechanisms of resistance (EFF, RPP, enzymatic inactivation for tetracyclines, and only enzymatic inactivation for beta-lactams). In addition, not all beta-lactamase classes have the same phenotypic evolutionary pathways (class C beta-

lactamases/cephalosporinases barely evolved phenotypically; PMID: 11959544, and other works by Barlow-M and Hall-BG). Also, emergence can be related to the success of MGEs (e.g. PMID:20107608). This discussion should be revised.

(Remarks on code availability)

Reviewer #2

(Remarks to the Author)

This manuscript addresses the factors that have shaped the tetracycline resistome. Although the manuscript is well written and easy to follow, the motivation behind the actual question being asked (and the hypothesis that is mentioned in L124-134) is rather weak, and in the end, I was not sure what the authors were after. Most importantly, the relevance of the experimental set-up for a natural situation is questionable (see comments below). I agree with the authors that the bar-coded method used to study competitions is useful and can be expanded to study even more complex systems.

Below are specific comments on the manuscript

1. The use of the term 'fitness' throughout the manuscript is confusing and inconsistent. In L157 and L400-419, authors use 'fitness' to describe levels of resistance. Level of resistance is only one of many components of fitness, and hence the authors should probably not use fitness to describe these observations (since fitness in the traditional sense is not being measured). Again, in L129 authors talk about fitness costs, which I think is in regard to reduction in growth rates. Authors should define what they mean by fitness, and use the term in a consistent manner.

2. L136-L145 I think the authors need to discuss how the expression of these resistance genes in naturally occurring systems compares to the artificial expression system used by the authors. The use of a single expression system for all the mechanisms is understandable from an experimental point of view but it does not make sense if one wants to understand what is going on during evolution of a natural system. Thus, any conclusions regarding the selection dynamics of these mechanisms would be strongly influenced by how these genes are expressed and regulated over time and in response to various conditions and inducers (including levels of tetracyclines). For example, if any of these genes are induced only when the antibiotic is present, the reduction of growth rate in the absence of the drug (Fig. 2C) will have less bearing on the evolutionary outcomes. Similarly, the shape of the induction curve for each gene in response to varying concentrations of the different tetracyclines, and how different levels of enzymes affect both MIC and fitness, would be key to understanding the selection process.

3. L213-223 The enrichment of genes representing different resistance mechanisms when in a mixture in the absence of any drug does not follow the reduction in growth rate observed in Fig. 2C (i.e. high costs observed for DES1 and RPP). This is confusing and the authors should comment on why these two observations do not match.

4. L264-L276 Although the dynamics between the mechanism-specific mixes and the ALL mix are similar, the rates seem to be very different in the figures. It might be useful to quantitatively compare these dynamics.

Along similar lines, the authors should comment on whether DES1 and DES2 are secreted in the media in any form (I think they are periplasmic), which would then result in interaction between the different mechanisms.

5. L277-L302 This is an important conclusion, but I think this will depend on the growth rate of the strain at that particular antibiotic concentration, which is not discussed in this section. Perhaps adding dosage curves for these mutants at the same concentrations that were used in other experiments will be useful for the conclusions from this section.

6. L314-316 The difference in increase between EFF and RPP in tetracycline is not clear (Fig. 5A). If this was quantitatively compared, the statistics should be mentioned here.

7. L303-327 In the experiments where the ALL mix is transferred from one antibiotic to another, small amounts of the first antibiotic will also get transferred. Since we know that extremely low concentrations of drugs can still enrich for resistance mutations it will be useful for the authors to comment on whether this is affecting their observations (even if the dilution factor for each transfer is 100-fold). This is especially important for Fig 5B, D, E, and F, where DES1 is maintained in the population despite the change in antibiotics.

8. L328-L345 The observation of the high prevalence of DES1 mechanisms in natural isolates (as well as of DES1 + RFF) is peculiar given the results of Fig. 2C (high costs for these mechanisms). It will be useful if the authors can comment on why these systems are widespread despite having fitness costs.

9. L156 full stop is missing at the end of the sentence,

10. L166 empty strain  empty vector strain.

(Remarks on code availability)

REVIEWER RESPONSE

Reviewer #1 (Remarks to the Author):

The manuscript by Blake et al. first describes how the selection of tetracycline resistance genes occurs through the selection of specific resistance mechanisms by tetracycline antibiotics of different generations. This resembles the introduction of such antibiotics in the therapeutical arsenal and the emergence of different genes/resistance mechanisms over time. The work may be relevant to understanding some aspects related to the ecology (distribution, redundancy) and evolution (exaptation, phenotypic evolution) of antibiotic resistance.

The predominance of some resistance mechanisms, such as tetracycline efflux pumps, is observed in resistomes of pristine environments, but the rationale beyond these findings has not been addressed. In addition to the novelty of the results, the experimental approach is highly innovative. The high-throughput barcode sequencing protocol (CompAReSeq: COMPetitive Antibiotic Resistance SEQuencing) that can discriminate and quantify genotypes in mixed cultures provides a valuable approach to testing the evolutionary pathways of multiple genes in microbial populations. Although the idea is not new, it is first applied to antibiotic resistance. The work is interesting, but the interpretation of some results should be revised. Some aspects of the experimental approach should be further detailed.

Some comments for the author's consideration appear below.

We thank Reviewer #1 for their kind assessment of our experimental results and approach, and their feedback for how to improve the interpretation of some results.

1. The tetracycline resistome. The number of known EFF and RPP genes is higher than that evaluated in this work (EFF: 30 vs. 6; RPP: 12 vs 6), while DS1 are allele variants of tet(X) (Pe.g., PMID: 27065405). The criteria for selecting the pool of tet genes tested are not provided. Note that tet genes (the sample analyzed here and the known pool) are neither equally prevalent nor have the same genetic context preference. While some genes are common in G+ve and G-ve taxa, others are associated with particular phylogenetic backgrounds, which could influence their prevalence and expression and, thus, some experimental results (see below).

We selected tet genes which have been extensively studied and are representative of each mechanism's overall sequence diversity (**Supp. Fig. 1B**) (**Lines 150-154**). In this revision, we have added an analysis of the taxonomic diversity of each gene (i.e. genera count, and proportion G+ve vs. G-ve) (**Supp. Fig. 1C**). Note: DES2 was not included in visualizations because these were identified by functional metagenomic selections, and have not been associated with specific taxa. Of the 583 genera from which tetracycline resistance genes have been identified, 339 are represented by the selected genes (**Supp. Fig. 1D**). Generally, the genes we used are found in the greatest number of genera for each mechanism. The exception is tetC, but this was not included

because of its sequence similarity with tetA which is the most well-studied EFF gene. Tetracycline resistance genes generally are more likely to be found in G-ve rather than G+ve taxa, but most of the genes we selected are found in both (**Supp. Fig. 1C**) (**Lines 154-157**). Therefore, although we acknowledge that many tetracycline resistance genes are associated with particular taxa and that the genes we used do not exhaustively cover all known sequence variants in the tetracycline resistome, we conclude that the genes used are a representative subset of the overall sequence and taxonomic diversity. We have added to the Discussion (**Lines 423-429**) on this limitation and opportunity to apply our methodology for other resistome variants.

2. Clonal background. Using a single clonal background is a limitation of the work (also recognized by the authors). The burden of the experimental work is impressive, and the possibility of extending the analysis to other strains, either the same or different species, seems unfeasible. However, the reader misses a more profound discussion regarding particular genes. EFF genes analyzed include some exclusively detected in G-ves (A, B, E, and G) and others widely distributed in G+ves (L, 39). The last ones were not predominantly selected in the experiments. Conversely, RPP genes are common in G+ve and G-ve or confined to G+ve taxa. Your results show a selection bias towards those with a broader distribution. A MIC assay of these constructions in other clonal backgrounds will be welcome to discard weird discussions about MICs. This value could change depending on the taxa background.

We appreciate the reviewer's kind comments about the scope and scale of our experimental work. We agree that testing fitness costs across strains would be valuable in follow up work and also agree with the reviewer that doing so is outside the scope of this manuscript. To highlight the importance of this consideration, we have added a discussion of a previous report, Jiang et al. 2021, which evaluated tetracycline resistance genes in different host contexts (**Lines 416-423**). To further acknowledge this important point about strain background context, we have made sure to go through our manuscript to verify that we qualify our claims about the fitness costs of the tetracycline resistome as assessed by expression in *E. coli* (**Lines 23, 148, 163, 217, 271, 296, 365-370, 392, 414-415, 453**).

3. Epidemiological data could enrich the discussion about the parallelism between introducing each antibiotic class in the therapeutic arsenal and the emergence of each resistance mechanism. The temporal sequence applies to G-ve and G+ves bacterial species where tetracycline resistance is extremely common (PMID: 25088811).

We have added discussion of the increasing number of tetracycline resistance genes, increasing abundance of these genes, and their increasing taxonomic breadth over time (**Lines 116-123**). This includes citations to review articles from 1996, 2001, and 2005 which describe the number of new species which tetracycline resistance genes had been found in. We have also included

Discussion on the paper referenced by the reviewer (**Lines 384-389**). Additionally, we generated a timeline figure illustrating how the introduction with each tetracycline generation coincides with the emergence of new resistance mechanisms in human pathogens (**Fig. 1A**), and a figure showing the increasing number of known resistance genes for each class over time (**Supp. Fig. 1A**).

4. Parallelism between the selection of tetracyclines and beta-lactams. The authors compare the sequential selection of tetracycline-resistant genes (which are associated with EFF, RPP, and enzymatic inactivation) with some beta-lactams (all linked to enzymatic inactivation and restricted to class A and class B beta-lactamases). This is a false equivalence because of different mechanisms of resistance (EFF, RPP, enzymatic inactivation for tetracyclines, and only enzymatic inactivation for beta-lactams). In addition, not all beta-lactamase classes have the same phenotypic evolutionary pathways (class C beta-lactamases/cephalosporinases barely evolved phenotypically; PMID: 11959544, and other works by Barlow-M and Hall-BG). Also, emergence can be related to the success of MGEs (e.g. PMID:20107608). This discussion should be revised.

We agree with the reviewer that the two are not equivalent; however, we believe that the parallels are numerous and important enough that they suggest our work may be representative of broader trends rather than being confined to the tetracycline resistome. To avoid the unintended appearance of an argument for equivalency, the parts of the discussion describing historical similarities to beta-lactam resistance has been pared down and we have more greatly emphasized the differences (**Lines 470-488**).

Reviewer #2 (Remarks to the Author):

This manuscript addresses the factors that have shaped the tetracycline resistome. Although the manuscript is well written and easy to follow, the motivation behind the actual question being asked (and the hypothesis that is mentioned in L124-134) is rather weak, and in the end, I was not sure what the authors were after. Most importantly, the relevance of the experimental set-up for a natural situation is questionable (see comments below). I agree with the authors that the bar-coded method used to study competitions is useful and can be expanded to study even more complex systems.

We thank Reviewer #2 for their kind assessment of our barcode sequencing method, and appreciate their feedback for how to strengthen the hypothesis and justify the experimental setup.

Below are specific comments on the manuscript

1. The use of the term ‘fitness’ throughout the manuscript is confusing and inconsistent. In L157 and L400-419, authors use ‘fitness’ to describe levels of resistance. Level of resistance is only one of many components of fitness, and hence the authors should probably not use fitness to describe these observations (since fitness in the traditional sense is not being measured). Again, in L129 authors talk about fitness costs, which I think is in regard to reduction in growth rates. Authors should define what they mean by fitness, and use the term in a consistent manner.

We have revised the manuscript to emphasize that we are evaluating each strain’s relative fitness—that is, their ability to reproduce and expand in mixed-culture populations compared to the other strains—rather than absolute fitness (Lines 32-33, 140). Accordingly, rather than referring to “fitness benefits” (i.e. increased MIC) and “fitness costs” (i.e. decreased growth rate), we instead refer to these genes as providing “benefits” and incurring “costs,” respectively (Lines 50, 139, 162, 173, 184), and emphasize that these contribute to fitness but are not the sole metric involved (Lines 422-423, 449-453).

2. L136-L145 I think the authors need to discuss how the expression of these resistance genes in naturally occurring systems compares to the artificial expression system used by the authors. The use of a single expression system for all the mechanisms is understandable from an experimental point of view but it does not make sense if one wants to understand what is going on during evolution of a natural system. Thus, any conclusions regarding the selection dynamics of these mechanisms would be strongly influenced by how these genes are expressed and regulated over time and in response to various conditions and inducers (including levels of tetracyclines). For example, if any of these genes are induced only when the antibiotic is present, the reduction of growth rate in the absence of the drug (Fig. 2C) will have less bearing on the evolutionary outcomes. Similarly, the shape of the induction curve for each gene in response to varying

concentrations of the different tetracyclines, and how different levels of enzymes affect both MIC and fitness, would be key to understanding the selection process.

We agree with the reviewer about including discussion of the importance of expression and regulation in natural systems. It is especially important for EFF genes, which are frequently observed to be controlled via the *tetR* regulator. We have added a discussion of this to **Lines 432-445**. Additionally, we have added an induction curve showing how in our IPTG-inducible system different levels of protein expression are correlated with higher/lower MIC (**Supp. Fig. 2B**). While acknowledging these limitations, we nevertheless stress the importance of using a controlled isogenic strain context for the purpose of this study to minimize variables. We recognize that the complexities of selective pressures and microbial growth dynamics will be much more complicated in the real world. However, we feel that using this controlled isogenic approach is essential for isolating the selective pressures of the 1-3 generation tetracyclines in a fair comparison.

3. L213-223 The enrichment of genes representing different resistance mechanisms when in a mixture in the absence of any drug does not follow the reduction in growth rate observed in Fig. 2C (i.e. high costs observed for DES1 and RPP). This is confusing and the authors should comment on why these two observations do not match.

As the reviewer has noted, DES1 and RPP genes had lower growth rates in monoculture, leading to an expectation that their relative abundances in the tetracycline-free mixed-culture experiment would decrease as well. We conclude that the contravening data showing their relative abundances remaining the same (while DES2 increased and EFF decreased) to be evidence that the degree of decreased growth rate was not large enough to be observable at the timescales and/or relative cell densities employed in the competition experiment, and acknowledge this point at **Lines 236-238**.

4. L264-L276 Although the dynamics between the mechanism-specific mixes and the ALL mix are similar, the rates seem to be very different in the figures. It might be useful to quantitatively compare these dynamics.

The different rates between the mechanism-specific and ALL mix are explained by the different initial relative abundances of each strain in each mix. For example, in the 4 $\mu\text{g/mL}$ tetracycline selection, the *tetB* strain increased +923% its starting relative abundance in the ALL mix and +228% in the EFF-specific mix. This 4x greater increase in the ALL mix coincides with *tetB*'s starting relative abundance being 4x lower than in the EFF-specific mix (4.5% vs. 16.6%).

4b. Along similar lines, the authors should comment on whether DES1 and DES2 are secreted in the media in any form (I think they are periplasmic), which would then result in interaction between the different mechanisms.

It has not been empirically determined whether or not tetracycline-inactivating enzymes are secreted; however, their requirement for NADPH to reduce the flavin cofactor during enzyme turnover likely means they are only functional in the cytoplasm. We have indicated as such in the introduction (**Lines 98-101**).

5. L277-L302 This is an important conclusion, but I think this will depend on the growth rate of the strain at that particular antibiotic concentration, which is not discussed in this section. Perhaps adding dosage curves for these mutants at the same concentrations that were used in other experiments will be useful for the conclusions from this section.

We agree that quantifying the growth rate of each strain in the context of the different antibiotics and concentrations would be useful evidence for this conclusion. We predict that there would be a direct relationship between the results of this experiment and each strain's change in relative abundance during the mixed-culture competitions. I.e. strains whose relative abundance increased the most in a given antibiotic-concentration would have the highest growth rate, and vice-versa. We have generated dosage curves for each strain at each drug concentration from our AST experimental data, showing the endpoint OD600 of each replicate in each drug-concentration tested, in **Supp. Fig. 3**.

6. L314-316 The difference in increase between EFF and RPP in tetracycline is not clear (Fig. 5A). If this was quantitatively compared, the statistics should be mentioned here.

There was no significant differences in the percent increase in relative abundance for EFF and RPP in tetracycline in this experiment (paired t-test).

7. L303-327 In the experiments where the ALL mix is transferred from one antibiotic to another, small amounts of the first antibiotic will also get transferred. Since we know that extremely low concentrations of drugs can still enrich for resistance mutations it will be useful for the authors to comment on whether this is affecting their observations (even if the dilution factor for each transfer is 100-fold). This is especially important for Fig 5B, D, E, and F, where DES1 is maintained in the population despite the change in antibiotics.

We have made note that very low concentrations of antibiotic will be transferred from experiment to experiment, and that this will amplify the "head start" that DES1s have over EFF and RPP strains following tigeicycline selection (**Lines 340-345**).

8. L328-L345 The observation of the high prevalence of DES1 mechanisms in natural isolates (as well as of DES1 + RFF) is peculiar given the results of Fig. 2C (high costs for these mechanisms). It will be useful if the authors can comment on why these systems are widespread despite having fitness costs.

We speculate that the widespread prevalence of DES1, despite the fitness costs alluded to by the reviewer, is because they are the only mechanism capable of conferring resistance to 3rd-gen tetracyclines (**Lines 165-166**). However, we cannot exclude the possibility that many may silence expression when not needed, mitigating the fitness costs.

9. L156 full stop is missing at the end of the sentence,

Corrected.

10. L166 empty strain  empty vector strain.

Corrected.